# Comparison of Aerial and Ground 3D Point Clouds for Canopy Size Assessment in Precision Viticulture

Andrea Pagliai [1], Marco Ammoniaci [2], Daniele Sarri [1], Riccardo Lisci [1], Rita Perria [2], Marco Vieri [1], Mauro Eugenio Maria D'Arcangelo [2], Paolo Storchi [2] and Simon-Paolo Kartsiotis [2,*]

1   DAGRI—Department Agricultural, Food Production and Forest Management, University of Florence, Piazzale delle Cascine 15, 50144 Firenze, Italy; andrea.pagliai@unifi.it (A.P.); daniele.sarri@unifi.it (D.S.); riccardo.lisci@unifi.it (R.L.); marco.vieri@unifi.it (M.V.)
2   CREA—Council for Agricultural Research and Economics, Research Centre for Viticulture and Enology, Viale Santa Margherita 80, 52100 Arezzo, Italy; marco.ammoniaci@crea.gov.it (M.A.); rita.perria@crea.gov.it (R.P.); mauro.darcangelo@crea.gov.it (M.E.M.D.); paolo.storchi@crea.gov.it (P.S.)
*   Correspondence: simone.kartsiotis@gmail.com

**Abstract:** In precision viticulture, the intra-field spatial variability characterization is a crucial step to efficiently use natural resources by lowering the environmental impact. In recent years, technologies such as Unmanned Aerial Vehicles (UAVs), Mobile Laser Scanners (MLS), multispectral sensors, Mobile Apps (MA) and Structure from Motion (SfM) techniques enabled the possibility to characterize this variability with low efforts. The study aims to evaluate, compare and cross-validate the potentiality and the limits of several tools (UAV, MA, MLS) to assess the vine canopy size parameters (thickness, height, volume) by processing 3D point clouds. Three trials were carried out to test the different tools in a vineyard located in the Chianti Classico area (Tuscany, Italy). Each test was made of a UAV flight, an MLS scanning over the vineyard and a MA acquisition over 48 geo-referenced vines. The Leaf Area Index (LAI) were also assessed and taken as reference value. The results showed that the analyzed tools were able to correctly discriminate between zones with different canopy size characteristics. In particular, the $R^2$ between the canopy volumes acquired with the different tools was higher than 0.7, being the highest value of $R^2 = 0.78$ with a RMSE = 0.057 $m^3$ for the UAV vs. MLS comparison. The highest correlations were found between the height data, being the highest value of $R^2 = 0.86$ with a RMSE = 0.105 m for the MA vs. MLS comparison. For the thickness data, the correlations were weaker, being the lowest value of $R^2 = 0.48$ with a RMSE = 0.052 m for the UAV vs. MLS comparison. The correlation between the LAI and the canopy volumes was moderately strong for all the tools with the highest value of $R^2 = 0.74$ for the LAI vs. V_MLS data and the lowest value of $R^2 = 0.69$ for the LAI vs. V_UAV data.

**Keywords:** precision farming; vegetation index; remote sensing; sensor; vineyard; spatial variability; mobile app; UAV; LAI; LiDAR

## 1. Introduction

Site-specific crops management represents an essential improvement in efficiency and efficacy of the different labors, and its implementation has experienced significant development in the last decades, especially for field crops [1,2]. In particular, precision viticulture techniques are becoming necessary in a production context focused on achieving the best possible operating efficiency and reducing costs by paying attention to environmental sustainability [3].

Precision Agriculture (PA) is defined as an agricultural, forestry and livestock management based on the observation, measurement and response of the set of inter and intra-field quantitative and qualitative variables that act in agricultural productions [4]. Data collection by proximal or remote sensors is the first step for acting a precision agriculture approach [5]. Then, collected data are interpreted and evaluated by an agronomical point

of view (e.g., canopy vigor) to traduce them into manual implementations or into inputs for variable rate technology (VRT) machines, which can perform the prescribed actions in a semi-automatic or fully automatic way [6,7]. Many studies stated that the canopy size of *Vitis vinifera* L. is closely correlated with the amount of sunlight intercepted, i.e., the amount of carbon assimilated [8–10]. It is also an essential characteristic in assessing crop management and plant health and water use [11]. The vineyard spatial variability is mainly due to exposure, soil composition, soil tillage, micro-climate, and water availability [12–14]. All these characteristics directly affect morphological, physiological and productive responses. Among the primary affected vegetative and productive responses there are canopy vigor, leaf area index (LAI), canopy volume, yield, grape quality, which can be further influenced by the type of rootstock used [15]. In specialty crops, the canopy size measurement is the main practice to fulfil the variable-rate applications that can be performed with manual techniques or by digital sensing tools. Usually, the measurements are then converted into corresponding canopy indicators (e.g., Tree Row Volume, Leaf Area Index, Leaf Wall Area, Unit Canopy Row, Ellipsoid Volume Method) [16,17].

In assessing the canopy size, the most used traditional methodologies and tools are listed as following: empirical and non-destructive methods [18–20], direct and destructive methods [21], point quadrat [22], optical and radiation sensors methods [21,23,24]. However, these methods can be very time-consuming and can have several uncertainties.

Thanks to technological advancements, more efficient, more precise and quicker measuring methods have emerged in the last decade [25,26]. Ultrasonic sensors were employed in several researches to enhance variable-rate applications [27–30]. Some researchers used them in orchards and vineyards in a continuous way (non-discrete), others used image processing from RGB cameras or laser scanners that measured the canopy shape or volume for selective real-time spraying [28,31–33]. The ultrasonic sensors enable an accurate measurement of canopy width in spot areas of canopy plants using ultrasonic waves. With the 2D LiDAR (Light Detection And Ranging) sensors, significant advancements in canopy recognition were made. LiDAR technology uses laser beams to create a point cloud of the canopy at varying angular resolutions and aperture angles [25]. As a result, the canopy complete vertical profile may be rebuilt. Recently, Mendez highlighted the potential of 3D LiDAR technology for canopy reconstruction in citrus and stated the critical issues for information extraction due to the lack of commercial software which allows quick processing [34].

Recently, thanks to open-source satellites imaging (e.g., ESA Sentinel-2, NASA Landsat-8), a considerable quantity of geo-referenced datasets are available for free. However, the resolution of satellites images is not often sufficient to highlight vineyard spatial variability because of different types of soil tillage and canopy management that can invalidate the canopy vigor data [35–37]. Therefore, other technologies are necessary to point out the vineyard spatial variability. Among them, the most recent tools that can be used in viticulture for measuring the canopy size in a precise and fast way are: Unmanned Aerial Vehicles (UAVs) [26,38], Mobile Laser Scanners (MLS) [39], multispectral sensors [40], Mobile Apps (MA) (e.g., Viticanopy) [41] and Structure from Motion (SfM) photogrammetry techniques [42]. These tools, also used in combination with each other, permit the reconstruction of a 3D model of the vine that can be processed to assess the canopy size in terms of volume, LAI and vigor [26,38,39]. Several studies proved the advantages and the potential in viticulture of the canopy 3D reconstruction to evaluate the spatial variability of the vineyard, to estimate the yields, to optimize pesticide treatments, fertilizers and water use [43–48].

In light of the mentioned framework, this article aims to evaluate, compare, and cross-validate the potentials and limits of MA, MLS and UAV to assess the canopy size parameters and their variability within the vineyard by processing 3D point clouds. Moreover, it is emphasized that this study is not meant to validate the quantitative volume assessment but to compare different tools used to calculate volumes and understand their limitations and advantages.

## 2. Materials and Methods

Three surveys were carried out for data collection, namely:

1. Ground data acquisition with a smartphone and mobile apps (MA);
2. Ground data acquisition with a mobile laser scanner (MLS);
3. Aerial data acquisition with an unmanned aerial vehicle (UAV).

The MA data were first processed to generate a LAI map of the test vineyard and then to reconstruct 3D point clouds of the tested vines. The MLS data were directly processed to create a NDVI (Normalized Difference Vegetation Index) map, a NDRE (Normalized Difference Red Edge) map and a 3D point cloud of the scanned vineyard rows. The UAV data were preliminary processed to generate an RGB orthomosaic of the vineyard and then reconstruct its 3D point cloud using the same MA algorithm and processing workflow.

The point clouds were processed to assess the canopy size (i.e., height, thickness, volume) of the sampled vines and then to compare the results of the different tools.

The complete workflow for the different tools is shown in Figure 1.

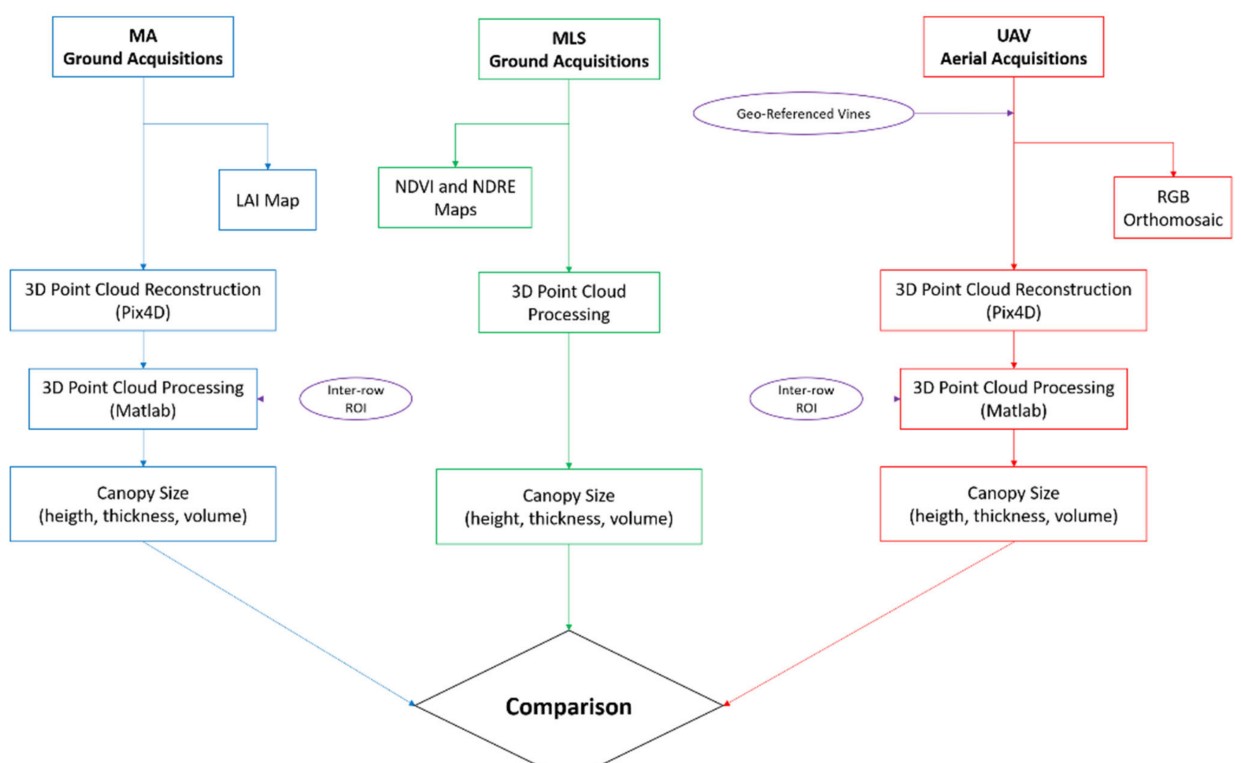

**Figure 1.** Overview of the methodology and general workflow. In blue, green and red, the steps for Mobile App (MA), Mobile Laser Scanner (MLS) and Unmanned Aerial Vehicle (UAV) data processing, respectively. In purple, the input data necessary for processing: the GNSS (Global Navigation Satellite System) position of the test vines was used to correctly geo-reference the orthomosaic, the inter-row and the Region of Interest (ROI) were used to select and process each test vine point cloud.

### 2.1. Experimental Site

Field tests were carried out in a vineyard in the Chianti Classico area, located in Gretole (43°27′23.0″ N; 11°13′51.9″ E), Castellina in Chianti, Siena, Italy (Figure 2). The experimental site was focused on 2 ha, where 48 test vines were sampled at three different phenological stages (BBCH 55, BBCH 65, BBCH 73) [49]. In each phenological stage, three types of measurements technologies, namely MA, MLS and UAV, were performed in the same day. The vineyard was located on a hillside, had a density of 5000 vines ha$^{-1}$ and the cultivar was the *Vitis vinifera* L. cv. 'Sangiovese'. The vines were 15-years old, trained with

a horizontal spur-cordon (4–6 buds per spur), at 0.80 m mean height from the ground with a planting distance of 2.50 × 0.80 m. The vine rows orientation was East-West.

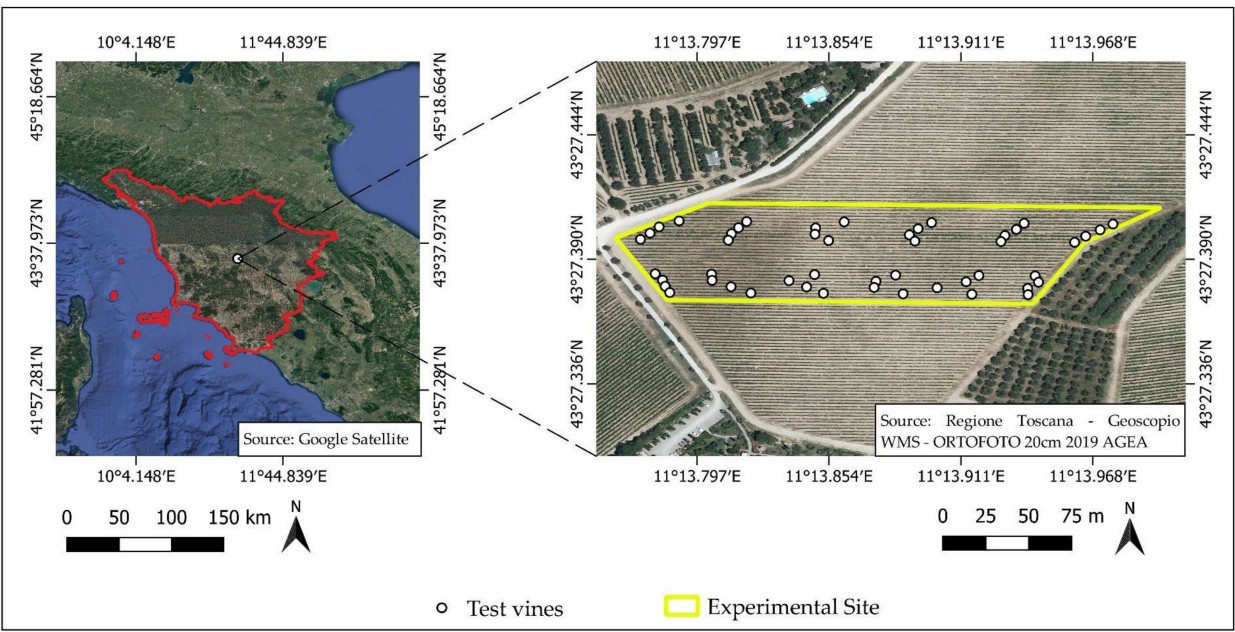

**Figure 2.** The geographical border of Tuscany is shown in red line, whereas the yellow line highlights the experimental site and the white points are the test vines locations.

### 2.2. Data Acquisition

Ground Measurements

- Leaf Area index

To characterize the spatial variability of the vineyard, the LAI was acquired over the 48 geo-referenced test vines using VitiCanopy, a free app for iOS and Android devices that has been developed by a team of researchers of the University of Adelaide and the University of Melbourne [41]. The working scheme of the app is reported in Figure 3.

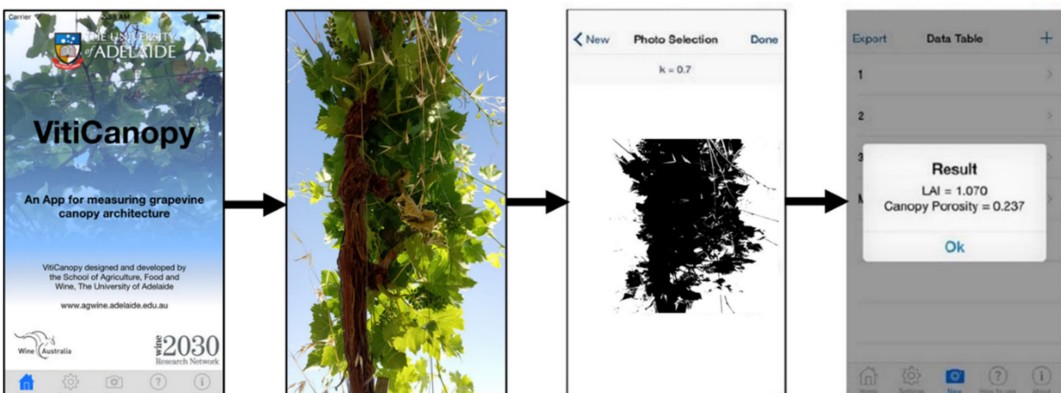

**Figure 3.** Example of Leaf Area Index (LAI) measurement procedure using the VitiCanopy app. From left to right, the app user interface, the shot image of the vine canopy, the segmented binary image of the canopy and the results in terms of LAI and canopy porosity.

VitiCanopy allowed to quickly and easily monitor the vine growth and the vigor of a vineyard and was used in place of traditional manual measures, which are time consuming (e.g., point quadrat method), not accurate and often require the destruction of the samples (e.g., defoliation of the vine to scan all the leaves and estimate the total leaf area).

The images were acquired using the frontal camera of an Apple iPad mini 2, placing it on the ground under the row line and at the middle of the vine cordon, following the indication provided by the app developer. Specifically, images were taken at 0.70 to 0.80 m from the cordon with a number of subdivisions of 5 (25 sub-images), a gap fraction threshold equal to 0.75 and a standard light extinction coefficient (k) = 0.7 [41].

- Mobile App (MA)

Ground images of the 48 geo-referenced test vines were collected in each survey date using a MA called Pix4Dcatch (Pix4D SA, Prilly, Switzerland), available for Android and iOS devices.

This app allowed ground-based 3D point clouds to be created using a smartphone or tablet camera. While the user scans a scene or an object, the app automatically records geo-referenced images with a very high overlap (i.e., 95%). Each vine was scanned on both sides by hand with approximately 150–200 images, as schematically shown in Figure 4.

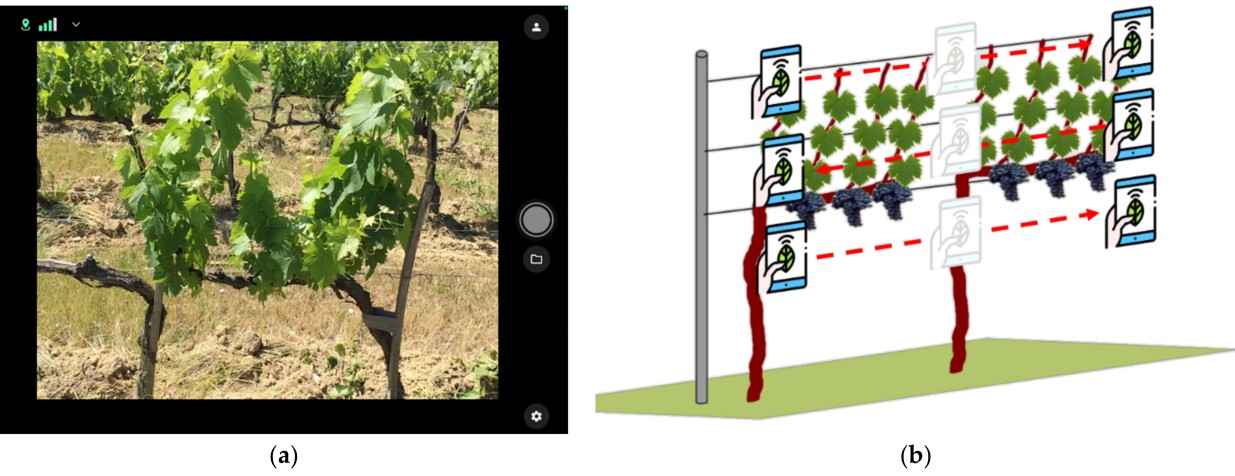

(**a**)                                                                          (**b**)

**Figure 4.** MA ground data acquisition: (**a**) Pix4Dcatch window app; (**b**) scanning pattern followed during the surveys.

The scans were performed by positioning the tablet perpendicular to the vine canopy at a distance of approximately 2 m. The acquisitions of the images were made by scanning from 0.30 m under the vine cordon and continuing in height on further two levels to reach the maximum height of 2.50 m above ground level in order to guarantee a good overlap between the photos. Half of the adjacent vines in the same row were also scanned to collect enough data for the sampled vine under study. The sampled vine was then extracted in the pre-processing phase and the borders cleaned.

- Mobile Laser Scanner (MLS) and Vigor Index

The MLS was carried out through a 2D LiDAR TIM 561 (Sick, Waldkirch, Germany) and a D-GNSS receiver (Differential—Global Navigation Satellite System) (AgLeader Technology, Ames, IO, USA) mounted on the rear part of a tractor (Figure 5a), where, on the right side of the Roll-Over Protective Structure (ROPS) the OptRx sensor is placed and coupled with the hardware and the rough book for data collection and storing, on the top of the front ROPS the D-GNSS receiver is placed and in the centre of the rear ROPS the LiDAR is oriented to the ground. In Figure 5b, the algorithm geometry scheme and the sensor are shown. In particular, LiDAR has an angular resolution of 0.33°, a working range from 0.05 m to 10 m, a scanning angle of 270° and a scanning frequency of 15 Hz.

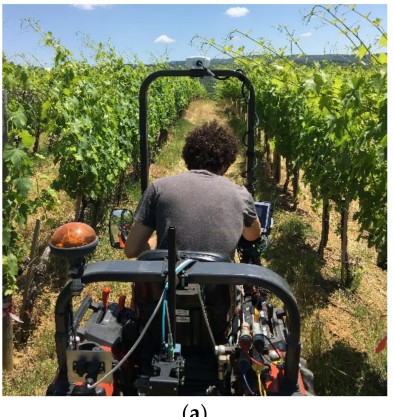

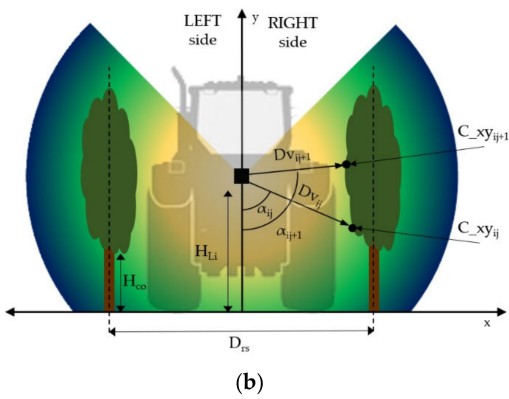

(**a**)  (**b**)

**Figure 5.** MLS ground data acquisition: (**a**) Main devices installed on the MLS for the canopy volume and vigour data acquisition. (**b**) LiDAR and algorithm working geometry. Model orientation and main geometries and measurements considered for the calculation of the volumes i.e., $H_{Li}$—LiDAR height from ground-level; $H_{co}$—cordon height from ground-level; $D_{rs}$—row-spacing; $Dv_{ij}$—distance between LiDAR and canopy at specified angular position *j* and at moment *i*, $\alpha_{ij}$—angle subtended by $DV_{ij}$, $C\_xy_{ij}$—pinpointed canopy data in cartesian coordinates.

Along with the LiDAR sensor, the OptRx (Ag Leader Technology, Ames, IO, USA) sensor was used to collect canopy vigor information. This proximal sensing tool was mounted in the central part of the tractor and arranged parallel to the vertical canopy axis, in order to avoid any noises of soil and grass. It measures the reflectance in the 630–685 nm (red), 695–750 nm (red edge) and 760–850 nm (NIR—Near InfraRed) wavebands, which were used to process the NDVI and NDRE indices. Reflectance data were collected simultaneously and at the same acquisition frequency of LiDAR data.

The average speed of MLS was 1.4 m s$^{-1}$ with an acquisition frequency of 5 Hz, in order to obtain a scan every 0.30 m.

- Unmanned Aerial Vehicle (UAV)

The DJI Phantom 3 Professional UAV (DJI, Shenzhen, China) (Figure 6a) was used as aerial platform. This UAV is equipped with a 12.4-megapixel CMOS sensor, has a diagonal size of 350 mm, a maximum take-off weight of 1280 g, a full flight time of 23 min, a top horizontal flight speed of 16 m s$^{-1}$ and maximum ascent and descent speeds of 5 and 3 m s$^{-1}$, respectively. The pitch angle range of the camera is approximately −90° to +30°.

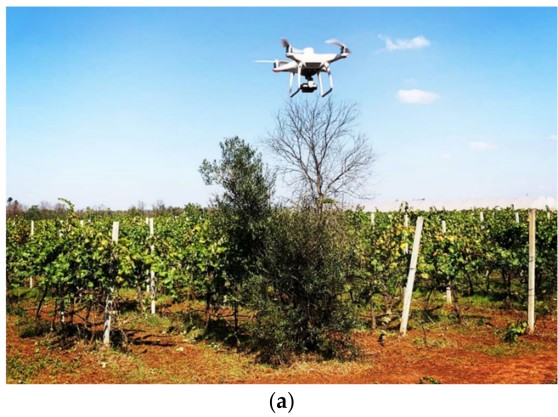

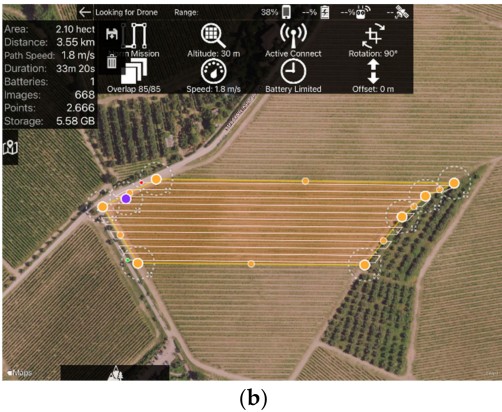

(**a**)  (**b**)

**Figure 6.** UAV aerial data acquisition: (**a**) DJI Phantom 3 Professional UAV in the vineyard; (**b**) UAV photogrammetry mission over the vineyard (Map Pilot app) with the overview of the interface for the flight mission and parameters. The yellow lines indicate the borders of the survey area, the green and red placemarks indicate the mission start and end points, respectively, and the purple placemark indicate the UAV take-off/landing point (i.e., the UAV pilot position).

On every survey date, the UAV imagery acquisition was made using the mobile app Map Pilot (Drones Made Easy, San Diego, CA, USA) (Figure 6b) to reconstruct the orthomosaic and the 3D point cloud of the vineyard in different phenological stages. UAV nadir, i.e., perpendicular to the terrain, photogrammetry images were acquired at a flight height of 30 m above ground level and a maximum cruise speed of 1.8 m s$^{-1}$. The overlap between two consecutive acquired images as well as the sidelap, i.e., the overlap between images in adjacent parallel flight lines, was 85%. The image resolution was 4000 pixels × 3000 pixels and the GSD (Ground Sampling Distance) was 1.3 cm/pixel. In order to guarantee a high quality of the post-processed orthomosaic, the "terrain following" feature was considered, i.e., the autopilot automatically adjusted the UAV altitude to keep the same relative height above the vineyard during the mission.

### 2.3. D Point Cloud Reconstruction

The 3D point cloud reconstruction was carried out by the software Pix4Dmapper Pro (Pix4D SA, Prilly, Switzerland). Pix4Dmapper Pro is a photogrammetric software that can quickly and automatically merge thousands of geo-referenced images to produce accurate orthomosaic, DSM (Digital Surface Model), point clouds and 3D models. It has been widely used in the fields of aerial photogrammetry and remote sensing applied to agriculture [50–52].

The software firstly evaluates the quality of the photogrammetric survey (e.g., good overlap between images), then marks key points between the images and automatically generates a densified point cloud and an orthomosaic of the test site or subject.

Pix4Dmapper Pro was used to:

(1) Generate three 3D point clouds of the test vineyard from the aerial RGB images, i.e., about 600 images for each UAV flight;

(2) Generate 144 3D point clouds of the test vines from the ground RGB images, i.e., about 200 images for each MA acquisition;

To geo-reference the aerial 3D point clouds in the WGS84 (World Geodetic System 1984) reference system, the position of 4 header poles located at the vertices of the test vineyard were measured with a GNSS RTK system. The CloudCompare v. 2.10.2 open-source software (http://www.cloudcompare.org/ (accessed on 3 February 2022) was used to remove the noise and separate the vines from the soil for the point clouds generated by the MA. In particular, the "segment" feature was used to manually select the points, or point cloud portions, that had to be removed.

The average density of the processed aerial 3D point clouds was about 1800 points m$^{-3}$, while the ground 3D point clouds had a density ranging between 50,000 points m$^{-3}$ and 500,000 points m$^{-3}$.

The point clouds generated by the MA and the UAV are reported in Figure 7, whereas it was not possible to represent graphically the MLS point clouds due to the different processing procedure (see Section 2.4).

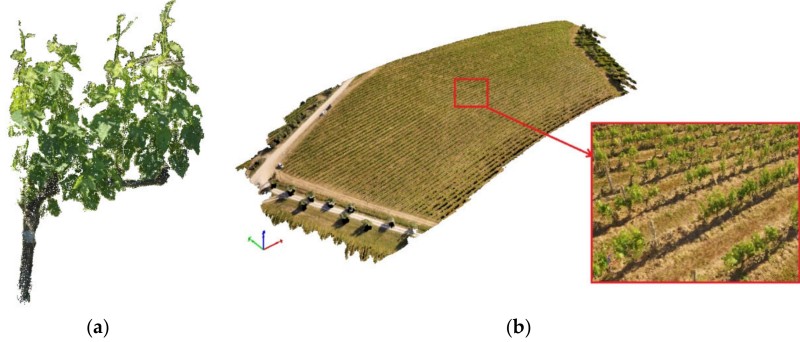

(**a**)        (**b**)

**Figure 7.** 3D Point Cloud Reconstruction: (**a**) MA vine point cloud processed with Pix4DMapper and cleaned with CloudCompare; (**b**) UAV vineyard point cloud processed with Pix4DMapper. In the detail, the vineyard rows and vines are shown.

### 2.4. 3D Point Cloud Processing Algorithm

The UAV and MA 3D point clouds were processed by an algorithm that was coded in Matlab (The MathWorks Inc., Natick, MA, USA). The algorithm was built following the approach defined in Comba et al. [26]. In particular, the 3D point cloud of a vineyard row portion, where the x, y and z axes were aligned with the vineyard row, the canopy width and the vertical axis, respectively, was processed through a series of spatial manipulation, also taking into account for the local soil slope. Then, the canopy density, height and thickness were calculated. For the canopy height and thickness assessment, respectively, the 80th percentile of the point cloud distribution projected in the xz plane and the difference between the 98th and the 2nd percentile of the point cloud distribution projected in the yz plane were found to be the best numerical descriptors with respect to the measured ones. For these reasons, the same descriptors were used in the present code.

Substantially, the code reads the processed UAV and MA 3D point clouds and gives as results the main canopy size parameters (i.e., thickness, height and volume). The working schemes of the algorithm for the UAV and MA 3D point clouds are reported in Figure 8.

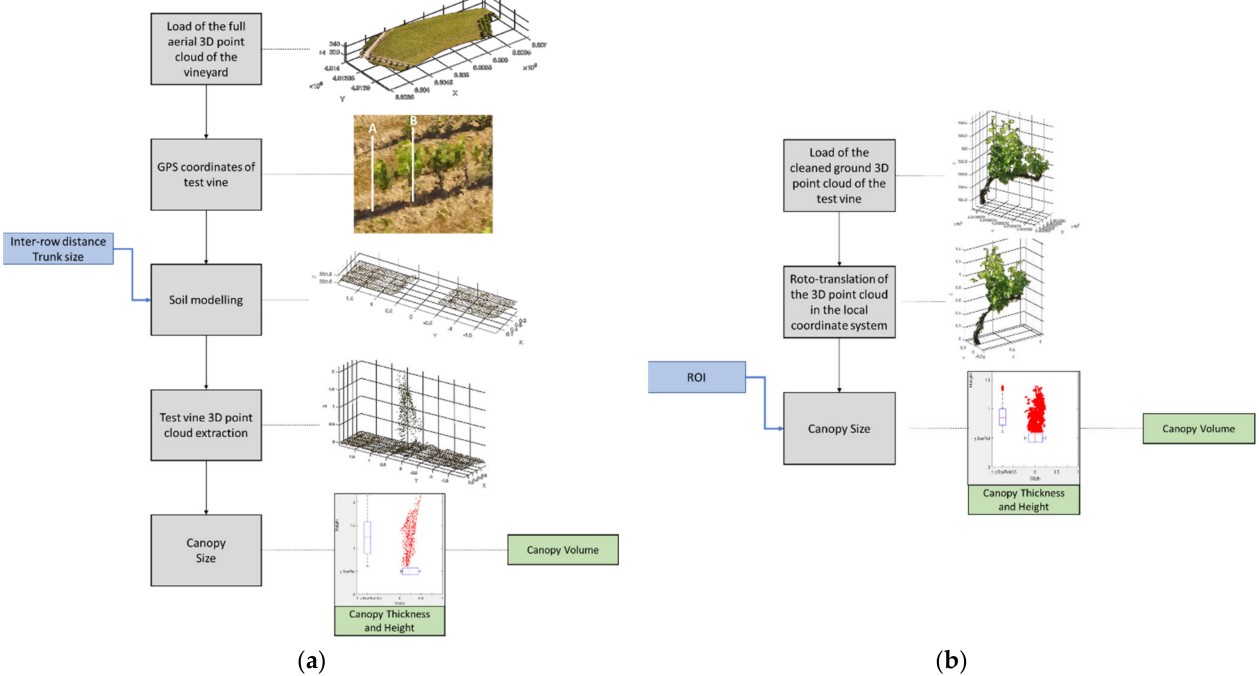

(**a**)                                                                                       (**b**)

**Figure 8.** Matlab algorithm processing scheme: (**a**) processing workflow for the aerial 3D point cloud; (**b**) processing workflow for the ground 3D point cloud.

For the UAV aerial survey (Figure 8a), the 3D point cloud was firstly loaded in its original coordinate system (WGS84), and the GNSS coordinates of the start (A) and end (B) of the trunk of the test vine were given as input to isolate it from the rest of the vineyard. Then, the algorithm took into account the soil slope in the proximity of the test vine, which is, in turn, roto-translated to achieve the local coordinate system origin in A and the x, y and z axes aligned with the vineyard row, the canopy width and the vertical axis, respectively. Finally, the canopy size was assessed in terms of thickness, height and volume.

For the MA ground survey (Figure 8b), the process was analogous, but in this case, the input 3D point cloud of the test vine was previously cleaned from noise and the soil was deleted thanks to CloudCompare. The roto-translation of the 3D point cloud of the test vine to match the x, y and z axes was carried out manually as well as the definition of the Region of Interest (ROI), i.e., the parallelepiped containing the selected test vine which was necessary to detect the processing canopy volume.

The MLS point clouds were processed separately by an integrated software. In particular, MLS data were processed, in real-time, by an algorithm that provided the canopy size parameters (volume, width, height) using canopy contours extraction operations, that consisted in converting the MLS raw data from polar to cartesian coordinates. Then, the contours widths (right and left) of the vertical canopy profiles were extracted for each MLS laser beam. After this step, the mean value of the widths was calculated and multiplied by the height of the canopy, extracted from the MLS data, for both sides, resulting in the total area of the canopy. These steps were repeated for each scan provided by the MLS during the work sessions. Lastly, the distance from one scan to another was calculated through the D-GNSS positioning and multiplied by the canopy areas previously calculated to achieve the total canopy volumes. The MLS-based algorithm was implemented in the software to automatically calculate the canopy volume. Such a process was repeated continuously throughout the survey stage. The software provided an output file (.csv—comma-separated values) with the data of the canopy volumes and their spatial position. Further information can be found in Pagliai et al. [53].

The canopy volume calculation for the three tools was carried out using the following equation:

$$V = T \cdot H \cdot L \tag{1}$$

where $T$ and $H$ are the canopy thickness and height, respectively, as calculated by processing the point clouds of the different tools, and $L$ was the cordon length that was considered equal to 1 m on average.

### 2.5. Data Analysis and Correlation

The LAI, NDVI, NDRE and the canopy size parameters, extracted by MLS, UAV and MA technologies, were analyzed using the open-source software R (R Core Team, 2021) [54]. The statistical analysis adopted to check the reliability and goodness of variables was the linear correlation between all measured and calculated parameters. The coefficient of determination ($R^2$) and the Root Mean Square Error (RMSE) were used to assess the model goodness and reliability. All the variables were checked to ensure a normal distribution of errors with the Shapiro-Wilk test ($p > 0.05$), by visual inspections (frequencies histogram, normal Q-Q plots and box plots) and by the verification of homoscedasticity using the Levene's test.

The "corrplot" package was used to visualize the $R^2$ data matrix and the "ggplot2" package was chosen to show the linear correlations and canopy parameters trends with scatters- and box-plots, respectively [55,56].

After the statistical analysis, the LAI and the canopy parameters were processed and transformed from punctual data into spatialized maps using the open-source software QGIS (https://www.qgis.org (accessed on 3 February 2022). The raster maps were generated using the IDW (Inverse Distance Weighting) interpolation algorithm [57] with the distance coefficient P equal to 5. Then, the maps were smoothed by the application of a Gaussian filter with a square grid of 9 × 9 pixels.

## 3. Results

### 3.1. Vineyard Spatial Variability Assessment

The spatial variability results in terms of LAI, NDVI and NDRE for each phenological stage are reported in Table 1. The LAI, NDVI and NDRE ranged in 0.34–3.11, 0.40–0.85 and 0.11–0.28, respectively.

**Table 1.** LAI, NDVI (Normalized Difference Vegetation Index) and NDRE (Normalized Difference Red Edge) values and percent coefficient of variation (C.V.%) over the three phenological stages.

| BBCH | Canopy Parameter | Max | Min | Mean | C.V.% |
|---|---|---|---|---|---|
| 55 | LAI | 0.99 | 0.34 | 0.60 | 23% |
| | NDVI | 0.65 | 0.40 | 0.57 | 9% |
| | NDRE | 0.18 | 0.11 | 0.15 | 13% |
| 65 | LAI | 2.02 | 0.47 | 1.10 | 21% |
| | NDVI | 0.78 | 0.55 | 0.70 | 7% |
| | NDRE | 0.23 | 0.15 | 0.20 | 10% |
| 73 | LAI | 3.11 | 0.89 | 1.93 | 25% |
| | NDVI | 0.85 | 0.64 | 0.78 | 6% |
| | NDRE | 0.28 | 0.18 | 0.24 | 13% |

In Figure 9, the LAI, NDVI and NDRE box-plots and the LAI zonation map in three classes are reported for each phenological stage, highlighting the trend of canopy growth in time and the intra-field spatial variability of the vineyard. The classes in the LAI maps were determined using the 25% quantile for the LOW class and the 75% for the HIGH class, being the MEDIUM class related to the 25–75% quantile interval.

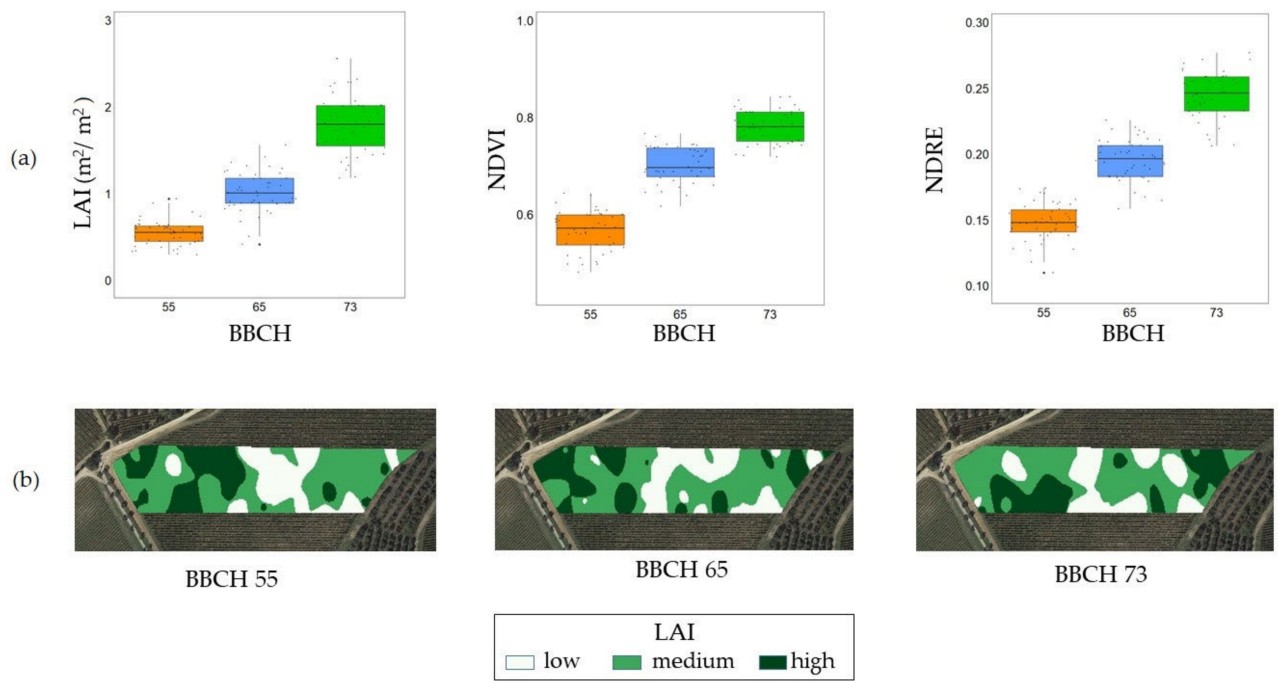

**Figure 9.** (**a**) LAI, NDVI and NDRE box-plots for the three phenological stages; (**b**) LAI zonation maps in three classes of the test vineyard for each phenological stage, where the classes were determined using the 25% quantile for the LOW class and the 75% for the HIGH class, being the MEDIUM class related to the 25–75% quantile interval.

The LAI, NDVI and NDRE data were used as reference values to validate the canopy size parameters reported in par. 4.2, since they are directly related to the amount of biomass [58], i.e., higher values of LAI, NDVI and NDRE represent denser vegetation areas and higher canopy volumes.

*3.2. Canopy Size Assessment*

The canopy size results in terms of thickness, height and volume of the 48 test vines for each phenological stage are summarized in Table 2, where the maximum, minimum and mean values are reported along with the percent coefficient of variations (C.V.%).

**Table 2.** Main canopy size results and percent coefficient of variation (C.V.%) in terms of thickness, height and volume for the different tools over the three phenological stages.

| BBCH | Value | Thickness | | | Height | | | Volume | | |
|------|-------|-----|------|------|------|------|------|------|------|------|
| | | UAV | MA | MLS | UAV | MA | MLS | UAV | MA | MLS |
| 55 | Max | 0.50 | 0.34 | 0.29 | 0.61 | 0.36 | 0.66 | 0.23 | 0.09 | 0.15 |
| | Min | 0.18 | 0.13 | 0.14 | 0.13 | 0.14 | 0.15 | 0.02 | 0.02 | 0.01 |
| | Mean | 0.29 | 0.21 | 0.21 | 0.40 | 0.24 | 0.42 | 0.12 | 0.05 | 0.09 |
| | C.V.% | 24% | 24% | 19% | 30% | 17% | 24% | 42% | 40% | 33% |
| 65 | Max | 0.61 | 0.45 | 0.35 | 1.05 | 0.70 | 0.97 | 0.48 | 0.20 | 0.34 |
| | Min | 0.28 | 0.21 | 0.20 | 0.28 | 0.29 | 0.40 | 0.12 | 0.04 | 0.08 |
| | Mean | 0.41 | 0.32 | 0.29 | 0.68 | 0.52 | 0.75 | 0.28 | 0.10 | 0.22 |
| | C.V.% | 20% | 19% | 10% | 24% | 19% | 16% | 29% | 40% | 23% |
| 73 | Max | 0.84 | 0.50 | 0.48 | 1.36 | 1.23 | 1.34 | 0.87 | 0.49 | 0.52 |
| | Min | 0.38 | 0.29 | 0.22 | 0.73 | 0.68 | 0.71 | 0.36 | 0.26 | 0.26 |
| | Mean | 0.58 | 0.40 | 0.36 | 1.07 | 0.94 | 1.04 | 0.59 | 0.38 | 0.40 |
| | C.V.% | 22% | 13% | 17% | 12% | 14% | 13% | 22% | 16% | 15% |

For the UAV, the thickness, height and volume ranged in 0.18–0.84 m, 0.13–1.36 m and 0.02–0.87 m$^3$, respectively. For the MA, the thickness, height and volume ranged in 0.13–0.50 m, 0.14–1.23 m and 0.02–0.49 m$^3$, respectively. For the MLS, the thickness, height and volume ranged in 0.14–0.48 m, 0.15–1.34 m and 0.01–0.52 m$^3$, respectively.

The UAV thickness estimation was higher than the other tools over the three phenological stages since the point cloud had more noise due to the inter-row (grass and soil) and to neighbor vines. In fact, the UAV point cloud is less precise in detecting a single vine than the other tools. On the other hand, the MA and MLS thickness estimations are very close one to each other. The height values are closer for the UAV-MLS comparison than for the UAV-MA and MLS-MA in all the phenological stages because of the more detailed point cloud generated from the MA processing with respect to the other ones. As results, the canopy volumes estimations are greater for the UAV due to a less precise detection of the single vine and a noisier point cloud, whereas the MA volumes estimations are the lowest among all the tools because of a more detailed and cleaner point cloud with respect to the UAV and MLS ones, being the latter coarser and with lower resolution.

The canopy thickness, height and volume data were processed and transformed into raster maps for each phenological stage using the same procedure that was described in par. 3.5. In Figures 10–12, the maps are reported along with the box-plots for each phenological stage and for each tool.

Linear regression models between each tool were analyzed to verify whether the different point clouds results were able to represent the canopy structure correctly. The results are shown in Figure 13, where H, T and V are, respectively, the canopy height, thickness and volume. As main result, it can be noted that the $R^2$ between the canopy volumes acquired with the different tools was higher than 0.7, being the highest value of $R^2 = 0.78$ with a RMSE = 0.057 m$^3$ for the UAV vs. MLS comparison, which indicates a strong correlation between them [59]. Such regressions indicate that all the tools have the same trends in representing the variability of the canopy volumes during the three phenological stages. The highest correlations were found between the height data for all the tools, being the $R^2$ values higher than 0.8 with the highest value of $R^2 = 0.86$ with a RMSE = 0.105 m for the MA vs. MLS comparison. For the thickness data, the correlations

were weaker, being the R$^2$ between 0.5 and 0.6 with the lowest value of R$^2$ = 0.48 with a RMSE = 0.052 m for the UAV vs. MLS comparison.

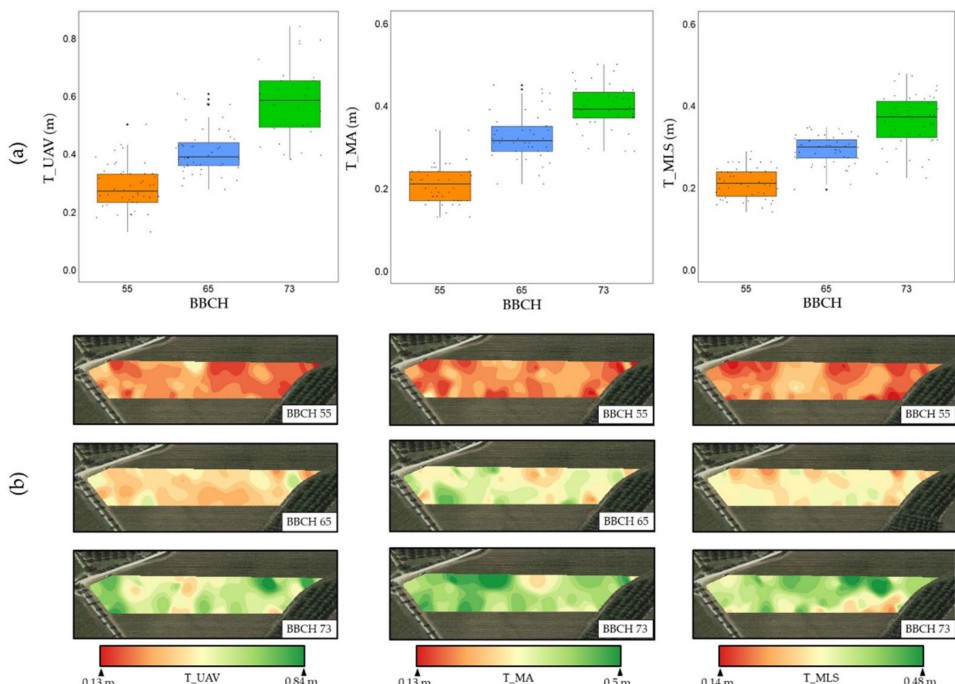

**Figure 10.** UAV, MA and MLS thickness results: (**a**) box-plots for the three phenological stages; (**b**) thickness zonation maps of the test vineyard for each phenological stage, where a single color scale was used starting from the minimum value (red) to reach the maximum value (green).

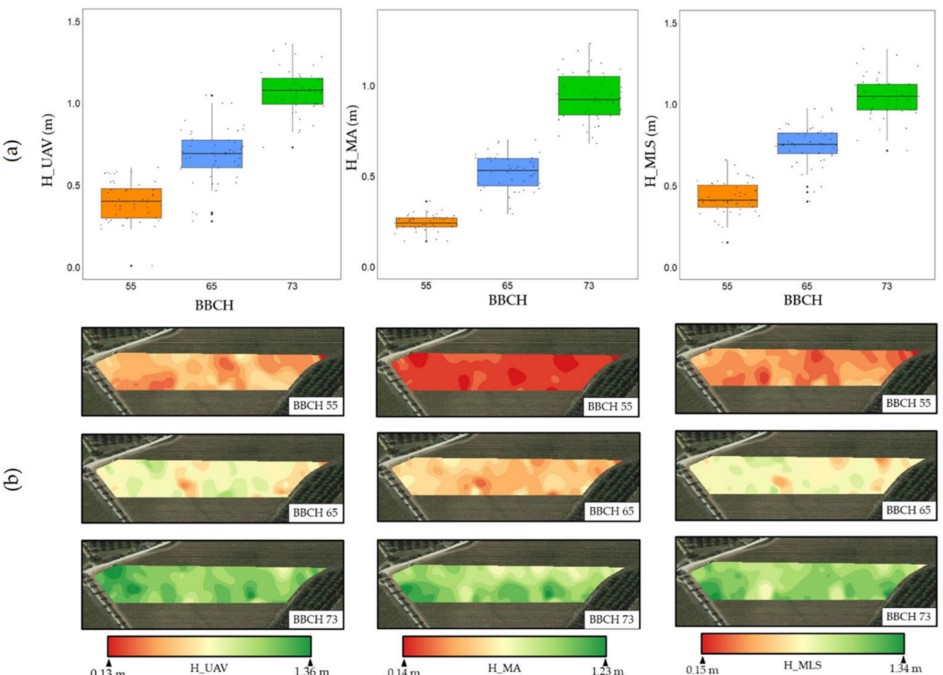

**Figure 11.** UAV, MA and MLS height results: (**a**) height box-plots for the three phenological stages; (**b**) height zonation maps of the test vineyard for each phenological stage, where a single color scale was used starting from the minimum value (red) to reach the maximum value (green).

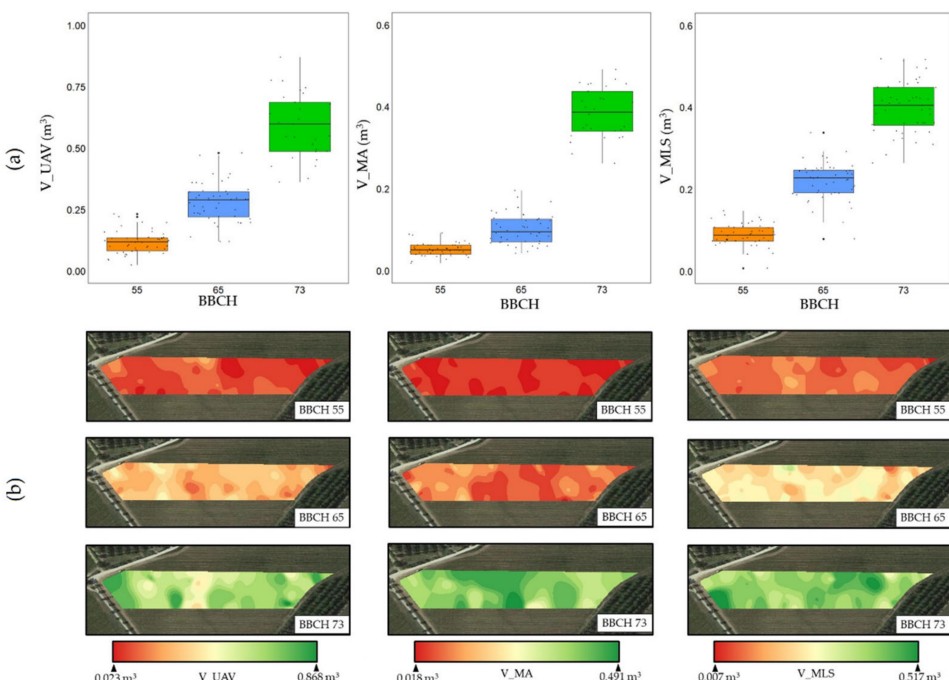

**Figure 12.** UAV, MA and MLS volume results: (**a**) volume box-plots for the three phenological stages; (**b**) volume zonation maps of the test vineyard for each phenological stage, where a single color scale was used starting from the minimum value (red) to reach the maximum value (green).

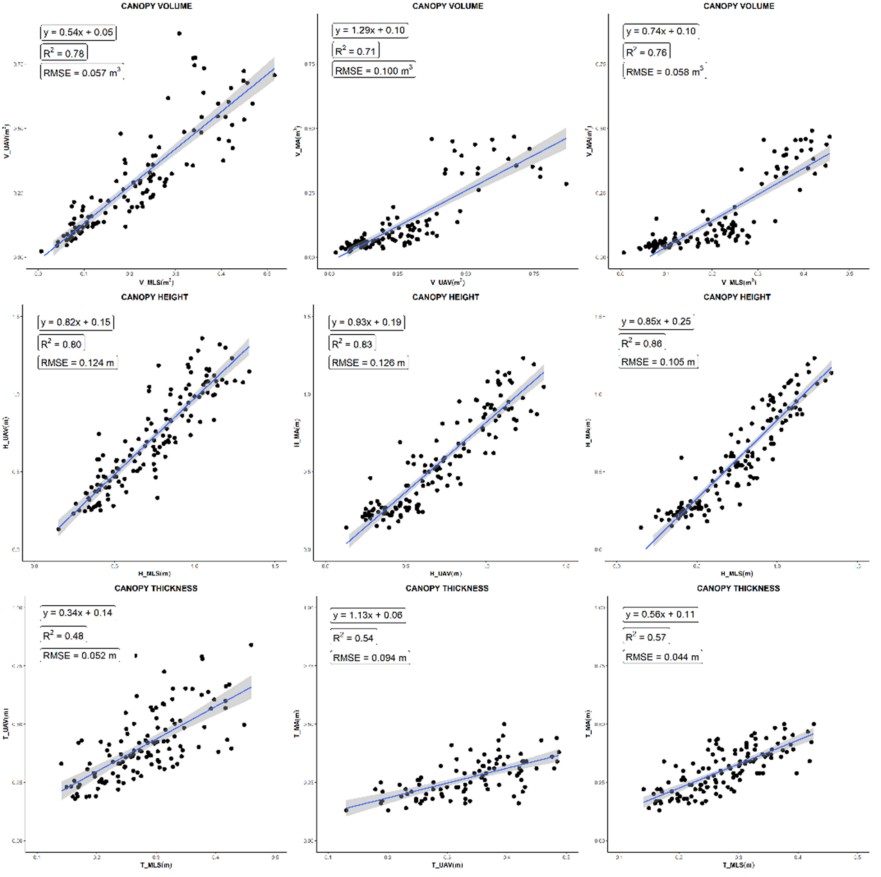

**Figure 13.** Scatter plots of the 48 sampled vines in the three phenological stages for all the tools.

The full $R^2$ matrix is reported in Figure 14, where it is shown that the correlation between the LAI values and the canopy volumes was moderately strong (>0.65) for all the tools. Being the measured LAI values taken as reference data to represent the test vineyard spatial variability, this indicates that all the point clouds were able to correctly represent the spatial variability of the canopy size in all the analyzed phenological stages, with the highest value of $R^2 = 0.74$ for the LAI vs. V_MLS data and the lowest value of $R^2 = 0.69$ for the LAI vs. V_UAV data. Furthermore, good correlations were found for the NDVI and NDRE variables with respect to the measured LAI ($R^2 = 0.67$ and $R^2 = 0.74$, respectively) and with respect to the canopy volumes, being the best value of $R^2 = 0.79$ for the NDRE vs. V_MLS comparison. Interesting correlations were also found between H_MA and H_UAV with respect to V_MLS ($R^2 = 0.87$ and $R^2 = 0.8$, respectively) and between H_MA and H_MLS with respect to V_UAV ($R^2 = 0.78$ and $R^2 = 0.7$, respectively).

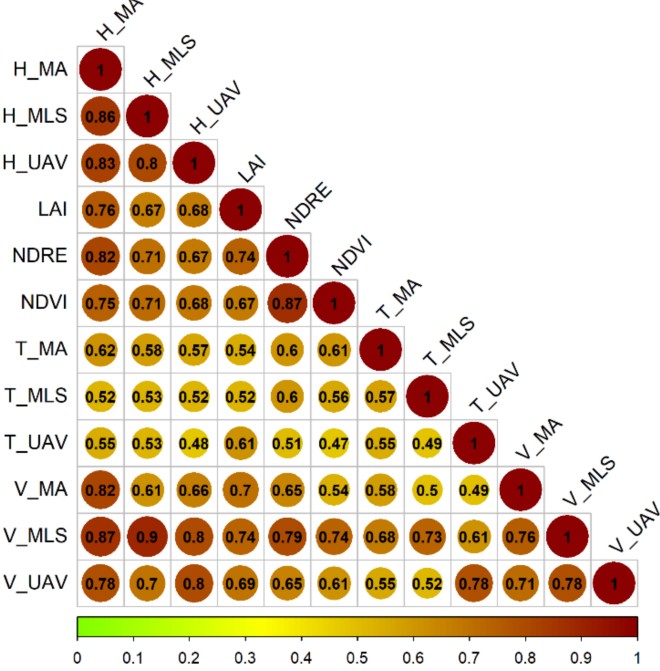

**Figure 14.** $R^2$ matrix for all the involved parameters and tools. H, T and V represents, respectively, the canopy height, thickness and volume.

## 4. Discussion

In this research, UAV, MA and MLS point clouds were compared to assess the canopy size parameters of vertical trained vines (*Vitis vinifera* L.). Manual measurements of the canopy volumes were not taken due to several uncertainties, such as the identification of the vine canopy boundary and being subjective and dependent on the person taking the measure itself as well as on the tool or strategy used to assess the canopy height and thickness [60,61]. Moreover, some researchers have found that manual measurements over-estimate the canopy thickness of about 30% with respect to LiDAR ones [62].

This study aims to give all the stakeholders an overview of the available tools and procedures used to assess the canopy size parameters of the vines in order to provide a reference for the vineyard precision management. The spatial variability detection is crucial in precision farming to automatize the VRT operations and to optimize the chemical inputs. Therefore, more than the precise quantitative estimations of the canopy volumes, it is essential to assess whether an area in the vineyard is more vigorous than another to differentiate the operations such as pruning, harvest, fertilization and crop protection stages, being the agronomist responsible to determine the quantitative applications based on the spatial variability maps.

Since LAI is directly related to the canopy volume in vineyards [28,29,63], LAI measurements computed with the app VitiCanopy were taken as an objective reference value to assess whether the different tools correctly characterize the canopy volume intra-field variability.

The results indicate that it is feasible to use 3D point clouds from the investigated tools to automatically compute the canopy height, thickness and volume of the vines and that the canopy size parameters variability in the test vineyard is detected correctly by all the tools in the analyzed phenological stages. The results highlight the possibility to use different tools to determine the vines canopy growth trend, thanks to a good correlation that was found between them for the same variable.

UAV technology was widely used to assess canopy volumes and it was shown to be a quick and low-cost solution compared to ground measurements of canopy size parameters [50,60,64–69]. The UAV point cloud processing led to similar results in canopy volume values concerning other research, but they used a voxel method [70] and an alpha-shape approach [38,71]. Therefore, direct comparison is not possible. Among different UAV-based measurements, canopy volume was found to be more sensitive to changes in canopy structure, compared to NDVI and projected canopy area, and demonstrated a more significant potential to assess the outcomes of a range of canopy management practices [72]. In addition, the MLS technology has also been widely used to assess canopy characteristics (canopy height, thickness and volume) but the goodness of canopy characterization was, mainly, compared to manual estimation. [28,61,62,64,73,74]. Rosell et al. [74] and Llorens et al. [28] stated that LiDAR is a valuable tool to characterize the canopy parameters and provides very precise canopy characterization. Instead, the MA technology based on the Pix4DCatch app is very recent (2020) and no research was found on its use in assessing vines canopy volumes. However, it is possible to affirm that this technology is a high-resolution photogrammetry solution to reconstruct high-detailed point clouds that is based on the same principle followed by the UAV technology. Even though the cost for the photogrammetric software to process the photos can be a bottleneck, an advantage of this method is the fact that it is not necessary to buy any expensive tool to take the photos nor an expensive hardware to process them (common smartphones and PC are enough), which makes it the most cost-effective and versatile approach for wineries with small vineyards scattered in distant areas. Moreover, the use of open-source photogrammetric software such as Open Drone Map (https://www.opendronemap.org/ (accessed on 3 February 2022) could overcome the economic limitation due to the software cost.

Confirmed that all the analyzed tools can assess the intra-field variability of the canopy size parameters, their advantages and limitations can be assessed. The UAV technology allows to quickly map lots of hectares, taking about 20 min of acquisition time for 2 ha of vineyard at an altitude of 30 m and 2 h of processing time with a standard laptop. For this reason, this solution can be more practical and economically relevant in medium-to-big wineries (>20 ha) and hilly environments. However, it requires trained staff and specific requirements to respond to national and international laws. The MA technology generates the more detailed point clouds but its measurement is punctual. It can be helpful for research purposes or in small wineries (<5 ha) because, at this time, the processing procedure requires a lot of work. However, if all the processing parts are automatized, it can become a powerful tool to directly assess the canopy volumes by the agronomists or farmers in order to support their vineyard management decisions more rationally. The MLS technology has the advantage to be an on-the-go system that can be installed directly on farm tractors so that the data are collected automatically during field operations. On the other hand, this technology is more time consuming than the UAV one, being 1.5 h the acquisition time for 2 ha, and the LiDAR sensor requires maintenance since it is very sensitive and susceptible to dust and its use can be complex in high-slope vineyards [75].

For all the tools, the processing procedures were time consuming and required enough computational resources to be performed in an efficient way. Some limitations and issues were experienced during the trials. In particular, the use of VitiCanopy to measure the

LAI experienced difficulties in early stage (BBCH 55) and in vines with a low canopy volume, because the cordon was detected as a part of the canopy volume itself, making an overestimation of the LAI. On the other hand, the Matlab algorithm used for the UAV and MA data overestimates the canopy thickness when the canopy is very well developed (i.e., BBCH 73). The overestimation is due to some very long branches that deflected inside the vineyard inter-rows so that the generated vines point clouds were much thicker than they actually were. For the MLS processing, the limitations are due to the data verification that has to be carried out manually and that the point cloud cannot be graphically visualized for further investigations. Another critical issue is the post-processing data analysis, how highlighted by Rosell et al., 2009 and Cheraiet et al., 2020. Despite these issues, in the last years many improvements regarding automated MLS data processing were developed and one of them was used in this study [53,75–77].

## 5. Conclusions

In this study, different digital tools, namely MA, MLS and UAV, were used to create 3D point clouds of test vines (*Vitis vinifera* L.) in order to assess the canopy size parameters such as thickness, height and volume in three different phenological stages. The tools were compared in terms of ability to detect and characterize the spatial variability of the vineyard in order to generate zonation maps useful for a precision farming management and VRT applications. Along with these measurements, the LAI, the NDVI and the NDRE indices were also assessed, being the LAI values taken as reference data to represent the vineyard spatial variability. The results indicated a good correlation between all the tools in terms of detecting the intra-field variability and the canopy size parameters. In particular, the $R^2$ between the canopy volumes acquired with the different tools is higher than 0.7, being the highest value of $R^2 = 0.78$ with a RMSE = 0.057 m$^3$ for the UAV vs. MLS comparison. The highest correlations were found between the height data for all the tools, being the $R^2$ values higher than 0.8 with the highest value of $R^2 = 0.86$ with a RMSE = 0.105 m for the MA vs. MLS comparison. For the thickness data, the correlations were weaker, being the $R^2$ between 0.5 and 0.6 with the lowest value of $R^2 = 0.48$ with a RMSE = 0.052 m for the UAV vs. MLS comparison. The correlation between the LAI values and the canopy volumes was moderately strong (>0.65) for all the tools with the highest value of $R^2 = 0.74$ for the LAI vs. V_MLS data and the lowest value of $R^2 = 0.69$ for the LAI vs. V_UAV data.

All the tested tools demonstrated to have some advantages and limitations: the UAV technology allows to quickly map lots of hectares but it requires trained staff and specific requirements to respond to national and international laws, the MA is a more punctual measurement but cheaper than the other tools so more affordable by small farms, the MLS can be installed directly on farm tractors so that the data can be collected automatically during field operations. The major limitations for all the tools are related to the data processing step which is time consuming and requires proper computational power.

Further developments of this study can be the use of UAV-based multispectral imagery, the automatization of the algorithms and the processing steps as well as the creation of prescription maps for pesticide treatments, based on the canopy volume maps.

**Author Contributions:** Conceptualization, S.-P.K. and A.P.; methodology, S.-P.K., A.P., D.S. and M.A.; software, S.-P.K. and R.L.; validation, S.-P.K., A.P., D.S. and M.A.; formal analysis, S.-P.K., A.P., D.S. and M.A.; investigation, S.-P.K., A.P., D.S., R.L. and M.A.; resources, R.P., D.S., M.E.M.D., P.S. and M.V.; data curation, S.-P.K., A.P. and M.A.; writing—original draft preparation, S.-P.K., A.P., R.P., D.S. and M.A.; writing—review and editing, S.-P.K., A.P., R.P., D.S. and M.A.; visualization, R.P., D.S., M.E.M.D., P.S. and M.V.; supervision, R.P., D.S., M.E.M.D., P.S. and M.V.; project administration, P.S. and M.V; funding acquisition, P.S. and M.V. All authors have read and agreed to the published version of the manuscript.

**Funding:** This research was funded by Regione Toscana (Italy) through "KATTIVO" Project, EU Rural development Program 2014–2020—PEI-AGRI, grant number 12927/2018.

**Acknowledgments:** The authors would like to thank the wineries Tenute Ruffino and Agricola San Felice that supported the experiments conducted above the test vineyard.

**Conflicts of Interest:** The authors declare no conflict of interest.

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
