# Peer review of "Comparison of Aerial and Ground 3D Point Clouds for Canopy Size Assessment in Precision Viticulture"

_remotesensing, doi:10.3390/rs14051145_

Round 1

Reviewer 1 Report

This is an interesting and easy-to-read paper about the use of different tools for point cloud generation and canopy parameters estimation, which is structured in a way that can be very practical for viticulture managers, specially to assess and map intra-parcel spatial variability. The comparative assessment is also useful from the scientific point of view. The paper is well structured, it has clear objectives, and results and discussion have practical interest.

In my opinion this manuscript is suitable for publication, once some minor corrections are done. I mention below some errors and typos that should be corrected before publication:

L-113: Define NDRE the first time

It is not clear the meaning/use of Inter-row ROI in Fig. 1

Describe better the caption of Fig. 2 (e.g., what is the red polygon, a regional limit?)

The caption of Fig. 5 doesn’t correspond to the figure

Fig. 8, it is not clear what is a) and b) in the figure

L-320: correct to “two indicators were used”

Consider deleting from L-319 to L-333, this is too basic to be described here, plus R2 and RMSE are mentioned before

Captions of Figs. 10, 11 and 12 do not correspond to the information of the figures. The platforms/tools and the canopy parameters are mixed up

L-511: It should be “Rosell et al.”

Thanks for your work.

Reviewer 2 Report

The manuscript presents a study that evaluates three approaches (Ground-based photogrammetry, UAV-based photogrammetry, and mobile laser scanner) for grapevine-related parameters extraction/estimation. Overall, it is a interesting study but there are some suggestions that should be followed to improve the manuscript:

Th authors should merge some paragraphs, rather than having multiple paragraphs composed of small sentences, most of them have no connection. This is notorious at the end of the Introduction and in Section 2 and Discussion.

Section 2 can be merged into the materials and methods.

The comparative analysis of the different approaches in the Discussion is good. However, it seems that a direct comparison of the obtained results with the ones obtained in similar studies is missing. Please include them in the revised version of the manuscript.

Other comments and suggestions in the attached PDF.

Round 2

Reviewer 2 Report

The authors improved the manuscript by addressing the comments and suggestions from the previous review.

Some minor concerns towards Figures 5 and 8. The legend of Figure 5 is too long; the authors can move part of it to the main text. Figure 8 seems not correctly formatted.

Regarding the point cloud noise and soil removal, describe with more detail which functions in CloudCompare were used to achieve that, if it was a manual procedure or if it was by using automated features included in the software (if so, describe them).

The statement in lines 511-513 is not completely true, despite Pix4Dcatch being a free app it steel requires a smartphone and, more importantly, photogrammetric software to process the acquired images (or a Pix4Dcloud license). Still, it is the most cost-effective solution among the three compared approaches.
